

# Effect of habitual physical activity on motor performance and prefrontal cortex activity during implicit motor learning

Fu-Miao Tan[1], Wei-Peng Teo[2], Jessie Siew-Pin Leuk[2] and Alicia M. Goodwill[2]

[1] School of Social Sciences, Department of Psychology, Nanyang Technological University, Singapore, Singapore
[2] Physical Education and Sports Science Academic Group, National Institute of Education, Nanyang Technological University, Singapore, Singapore

## ABSTRACT

**Background:** Acute bouts of exercise have been shown to improve motor learning. However, whether these benefits can be observed from habitual physical activity (PA) levels remains unclear and has important implications around PA guidelines to promote motor learning across the lifespan. This study investigated the effect of habitual PA levels on brain activity within the dorsolateral prefrontal cortex (DLPFC) during procedural motor skill acquisition.

**Methods:** Twenty-six right-handed healthy young adults had physical activity levels quantified by calculating the metabolic equivalent of task (METs) in minutes per week, derived from the International Physical Activity Questionnaire (IPAQ). Functional near-infrared spectroscopy (fNIRS) over the DLPFC was recorded to measure neural activation during a serial reaction time task (SRTT). Behavioural indicators of procedural motor skill acquisition were quantified as reaction time and accuracy of correct trials during the SRTT. DLPFC activation was characterised as task-related changes in oxyhaemoglobin ($\Delta[HbO_2]$).

**Results:** Findings showed that higher PA levels were associated with improvements in reaction time during procedural motor skill acquisition ($p = 0.03$). However, no significant effects of PA levels on accuracy or $\Delta[HbO_2]$ during procedural motor skill acquisition were observed. These findings show that while habitual PA may promote motor performance in young adults, this is not reflected by changes in the DLPFC area of the brain.

# INTRODUCTION

Motor learning comprises the foundation of physical development, motor performance for sports, recovery from neuromuscular injury, and maintenance of activities of daily living across the lifespan. Specifically, motor skill acquisition, also known as the "fast" or "early" component of motor learning, involves binding independent actions into cohesive motor chunks, and adjusting movements to error correct, and has been shown to display meaningful within-session performance improvements in motor execution speed and

Corresponding author
Alicia M. Goodwill,
alicia.goodwill@nie.edu.sg

accuracy (*Fitts et al., 1975*; *Karni et al., 1998*; *Dayan & Cohen, 2011*; *Doyon et al., 2018*). Emerging evidence has demonstrated that exercise may have positive benefits on the processes involved across all stages of motor learning (*Taubert, Villringer & Lehmann, 2015*), which could serve as a low-cost, accessible strategy for skill acquisition in elite sports, rehabilitation, and prevention of neuromuscular decline in ageing. For example, acute bouts of aerobic exercise performed before or after motor practice have been shown to improve motor skill acquisition and memory consolidation (*Roig et al., 2012*; *Statton et al., 2015*). Whilst acute exercise has shown promise to boost motor learning, these benefits may be dependent on the timing and intensity of the exercise (*Yanagisawa et al., 2010*; *Taubert, Villringer & Lehmann, 2015*; *Snow et al., 2016*; *Marin Bosch et al., 2020*).

The evidence for motor learning benefits from longer-term exercise interventions and/or habitual PA remains sparse, and associations have mainly been drawn from rehabilitation settings (*Quaney et al., 2009*; *Linder et al., 2019*). To date, studies in healthy older adults have shown positive associations between PA and motor skill acquisition in the upper extremities of older adults (*Hübner & Voelcker-Rehage, 2017*). In young healthy adults, negative associations between PA and reaction time were observed, concurrent with differences in pre-movement brain activity (*Cirillo, Finch & Anson, 2017*). As modern lifestyles and work demands have resulted in declining PA trends among younger adults (*Haskell et al., 2007*), understanding the effect of habitual PA on motor skill acquisition and underlying mechanisms is imperative to guide public health recommendations for PA to sustain brain health and motor function across the lifespan.

It is well established that exercise promotes the release of brain-derived neurotrophic factors (BDNF) (*Taubert, Villringer & Lehmann, 2015*), and insulin-like growth factor 1 (IGF-1) (*Carro et al., 2000*), down-regulates inhibitory neurotransmitter γ-aminobutyric acid (GABA) related genes (*Molteni, Ying & Gómez-Pinilla, 2002*) enhance long-term potentiation-like plasticity in motor learning-related brain regions (*Taubert, Villringer & Lehmann, 2015*). Moreover, PA levels have been positively associated with grey matter volume and executive function task-based brain activity in the prefrontal cortex (PFC) (*Northey et al., 2020*; *O'Brien, Kimmerly & Mekari, 2021*). Notably, the dorsolateral prefrontal cortex (DLPFC) has been shown to play an important role during the early phase of motor skill acquisition, particularly when motor performance is unrefined and the cognitive load is high (*Leff et al., 2008*). Current literature indicates that enhanced DLPFC activation during the initial phases of motor skill acquisition can be explained by evidence linking the DLPFC to visuospatial working memory (VSWM) (*Baddeley, 1986*; *Courtney et al., 1997*; *Leff et al., 2008*; *Lin et al., 2022*). The acquisition of novel motor skills requires individuals to hold, manipulate and monitor incoming information in the VSWM, resulting in increased attentional demand as they actively discover and acquire the new spatiotemporal arrangement during the task (*Leff et al., 2008*). Indeed, both the ventrolateral and dorsolateral prefrontal cortices have been shown to be actively involved in procedural motor sequence learning tasks (*Cao et al., 2022*; *Polskaia et al., 2023*). Taken together, it is plausible that lifestyle behaviours influencing DLPFC activity, such as PA, could affect the neural processes during procedural motor skill learning.

Therefore, this study utilised functional near-infrared spectroscopy (fNIRS) to examine whether levels of habitual PA differentiated implicit motor skill acquisition and associated DLPFC neural activity in young, healthy adults. fNIRS is a non-invasive neuroimaging technique that measures changes in oxygenated haemoglobin concentration ($[HbO_2]$) following neural activation within a cortical brain region, and is suitable for investigating brain function during motor tasks due to its portability and movement tolerability (*Pinti et al., 2020*). It was hypothesised that subjects with adequate PA levels would have better motor performance and neural activation within the DLPFC during sequence-specific learning of the serial reaction time task (SRTT).

## MATERIAL AND METHODS

### Participants

Twenty-six healthy young adults were recruited from the local University and general population. The sample size was based on a previous study examining PA levels on motor reaction time and electroencephalography measures (*Cirillo, Finch & Anson, 2017*). It was estimated that the minimum sample size to observe significant differences in reaction time and brain activity at a 0.05 alpha cut-off and 80% power was 24, with an additional 10% buffer for dropout/poor quality fNIRS signals. All participants had no known history of neurological or visual impairments or any musculoskeletal conditions in their dominant hand and were predominantly right-handed. All participants attended a single session at the Motor Behaviour Laboratory, National Institute of Education, Nanyang Technological University. During the session, participants' age, PA levels, body mass index (BMI), and education were recorded. Participants then performed a serial reaction time task (SRTT) to examine implicit procedural motor skill acquisition, whilst dorsolateral prefrontal cortex (DLPFC) neural activity was concurrently recorded *via* functional near-infrared spectroscopy (fNIRS). The study was approved by the Nanyang Technological University Institutional Review Board (IRB-2021-603) and carried out in accordance with the declaration of Helsinki. All participants provided written informed consent prior to conducting the session.

### Physical activity

Habitual levels of PA were measured by administering the International Physical Activity Questionnaire (IPAQ) long format. This questionnaire has shown acceptable levels of reliability and concurrent validity in reflecting long-term habitual PA patterns (*Craig et al., 2003*; *Helmerhorst et al., 2012*). Participants completed the entire questionnaire, but subsequent physical activity levels were quantified by calculating the metabolic equivalent of tasks (METs) in minutes per week (*Piercy et al., 2018*).

### Serial reaction time task (SRTT)

To quantify implicit procedural motor skill acquisition, participants performed a serial reaction time task (SRTT; EPrime 3.0, Psychology Software Tools, Pittsburgh, PA, USA). Participants were instructed to place the index, middle, ring, and little finger of their right hand over the buttons, with each finger being responsible for pressing the 1, 2, 3, and
**(A)**

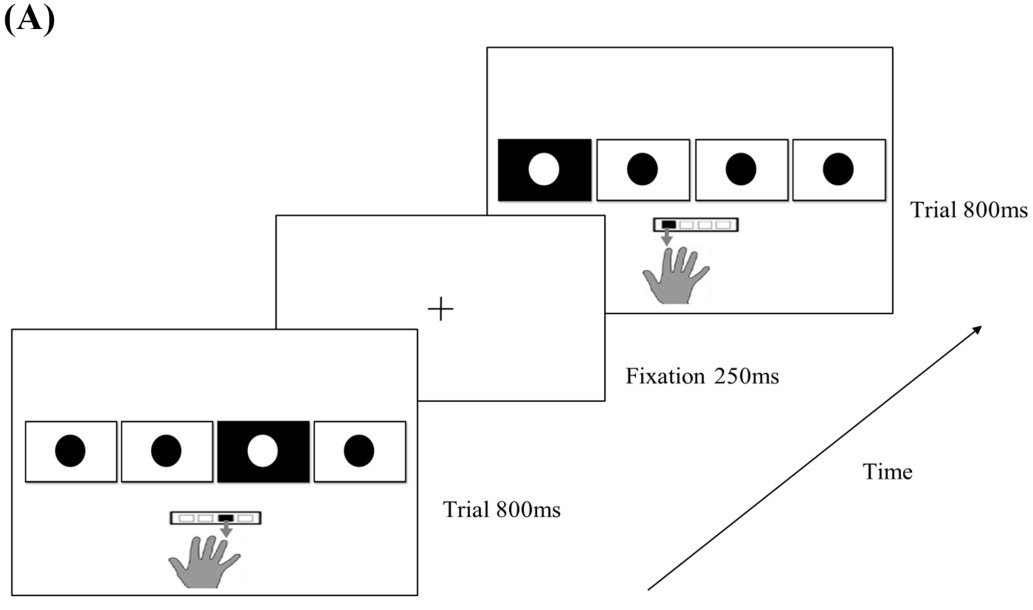

**(B)**

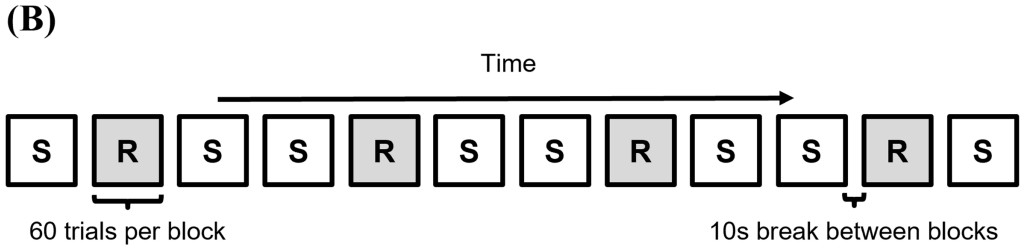

**(C)**

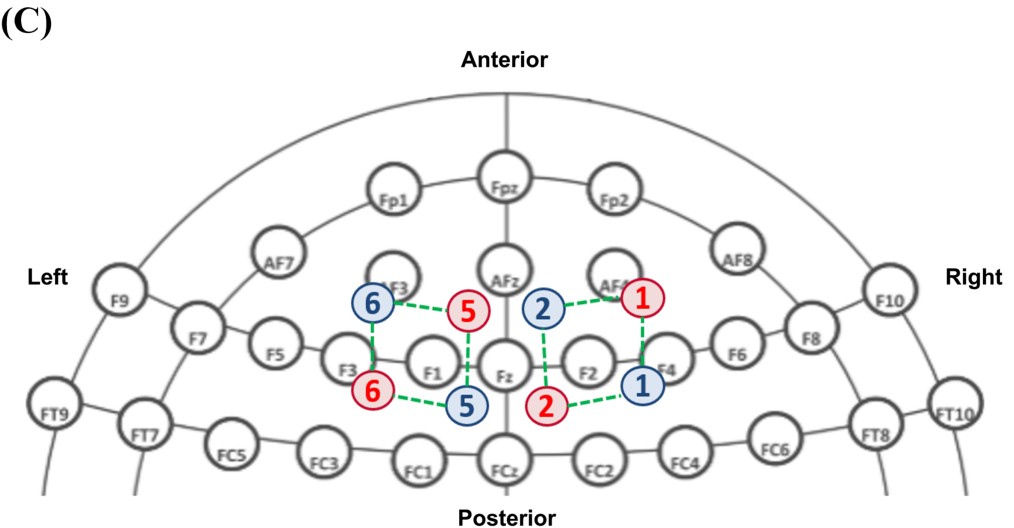

**Figure 1 Methods and materials utilised.** (A) Serial reaction time task (SRTT) trials across time; (B) pattern of sequence (S) and random (R) block presentation across the serial reaction time task; (C) infrared illuminating (red) and detecting optodes (blue) labelled with a corresponding number, with the channels (green), mapped onto dorsolateral prefrontal cortex (DLPFC) using the international 10–20

**Figure 1** (continued)
headspace. Note: 10-10 EEG system template used as background in (C) is derived from: https://www.
nature.com/articles/s41598-018-21716-z. The remaining parts are all original. The image of the hand
in (A) is drawn by the author #3 where she outlined her hand on a iPad before saving it as a image. The
rest of the components in (A) are built from the basic shapes provided in PowerPoint.

4 buttons on a standardised keyboard. The buttons correspond to the leftmost, left, right,
and rightmost square outlines (3 × 3 cm spaced apart by 1.5 cm) displayed horizontally on
a desktop monitor. Each trial began with a fixation cross that acted as a pre-cue. This was
followed by the visual cue showing four squares, with the target being filled in black and
overlain with a white dot (Fig. 1A).

Subjects responded to the visual cue by pressing the appropriate button as fast as
possible. The next trial proceeded after they pressed either the correct or incorrect buttons
or after the maximum stimulus cue time of 35 s had elapsed per block. RT and accuracy for
each trial were recorded. Individuals would initially familiarise themselves by performing
two randomly generated 12-element sequences, separated by a 10-s break in-between
(*Bhakuni & Mutha, 2015*). During the experimental session, trials were presented in either
a random, or deterministic pattern unbeknownst to them. The pre-defined repeating
sequence helped to capture procedural motor skill acquisition, whilst the embedding of
random trials between sequence trials ensured that learning was sequence-specific (*Nissen
& Bullemer, 1987*; *Robertson, 2007*; *Fitzroy et al., 2021*). Taken together, this provides a
specific and sensitive measure of visuomotor learning involving both cognitive and motor
processes by mapping visual cues to appropriate responses (*Robertson, 2007*). A total of
eight sequential (S) and four randomly (R) generated blocks (total of 12 blocks) were
presented in the order of S-R-S-S-R-S-S-R-S-S-R-S (sequence adapted from *Dumel et al.
(2016)*; Fig. 1B).

Each block contained 60 trials. A 12-element sequence of 2-1-4-3-4-1-2-3-1-3-2-4
(representing squares from left to right) (*Bhakuni & Mutha, 2015*), was repeated five times
per S-block (*Dumel et al., 2016, 2018*; *Cirillo, Finch & Anson, 2017*). The sequence was
bounded by the following constraints (*Witt, Ashe & Willingham, 2008*): (1) each square
would elicit a visual cue three times during one cycle of the sequence; (2) a cue was never
repeated consecutively (*e.g.*, 1-1-1-1); and (3) there were no complete thrills (*e.g.*, 1-3-1-3)
or runs (*e.g.*, 1-2-3-4 or 4-3-2-1). In each R-block, the visual cue location was determined
randomly by the software. The maximum time allocated to each block was 35 s, but only an
average of 26 s was needed to complete each block for all participants, with none exceeding
30 s. A 10-s break was given in between each block to allow for the hemodynamic response
to return to baseline levels, but participants were told to not lift their fingers off the keypad
and continue to fixate on the monitor display. Subjects took a 1-min break between the 6th
and 7th block, giving them time to adjust themselves and reduce the confound of fatigue.

Gain in declarative knowledge is known to inflate performance during similar motor
learning tasks (*Willingham, Nissen & Bullemer, 1989*; *Willingham & Goedert-Eschmann,
1999*; *Robertson, 2007*). To ensure that participants did not gain explicit awareness of the
sequence, they were verbally asked if they noticed a sequence pattern that was repeated

during the experiment. If implied so, subjects were told to list down their perceived sequence using the numbers 1 to 4 to contrast to the sequence aforementioned. Participants who correctly recalled five or more sequences consecutively were excluded from the study—as five is usually regarded as the approximate guessing rate (*Willingham & Goedert-Eschmann, 1999*; *Trofimova et al., 2020*). Ultimately, none of the participants were able to recall more than five consecutive sequences correctly.

## fNIRS data acquisition

A non-invasive, continuous, dual-wavelength fNIRS apparatus was used to quantify $\Delta[HbO_2]$ (µM) during the task (NIRS; Artinis Medical System, Oxymon MKIII, Zetten, Netherlands). fNIRS probe placement on the bilateral DLPFC followed the 10–20 international system (*Floyer-Lea & Matthews, 2005*; *Yanagisawa et al., 2010*; *Dayan & Cohen, 2011*; *Ohbayashi, 2021*; *Fitzroy et al., 2021*). The DLPFC coverage was confirmed using fNIRS optode location decider (fOLD), a MATLAB-based toolbox (The MathWorks, Inc., Natick, MA, USA) (*Zimeo Morais, Balardin & Sato, 2018*). The fNIRS montage consisted of four illuminating and detecting optodes arranged in an alternating fashion, with an inter-probe distance of 3 cm, resulting in a total of eight channels (Fig. 1C). Age differences for each participant were accounted for using the differential path-length factor (DPF) (*Scholkmann & Wolf, 2013*). Data collection was performed using the software Oxysoft 2.0.47 (Artinis Medical Systems, Elst, Utrecht, Netherlands), and sampled at a rate of 10 Hz, with wavelength intensities of 765 and 850 nm.

## fNIRS data processing

fNIRS data were processed using HOMER3, a MATLAB-based software (The MathWorks, Inc., Natick, MA, USA). Raw data was first visually inspected for the presence of a heartbeat, resulting in two participants being excluded for poor signal quality. Raw signals were first converted to optical density (OD) using the hmrIntensity2OD function. This was followed by automatic motion artefact tagging (hmrMotionArtifactByChannel; tMotion = 1.0, tMask = 1.0, STDEVthresh = 50.0, AMPthresh = 5.0), before being corrected using both spline interpolation (*Scholkmann et al., 2010*), and a Savitzky-Golay (S-G) filter (*Jahani et al., 2018*), (hmrMotionCorrectSplineSG; $p$ = 0.99, FrameSize_sec = 10, turnon = 1). After this, a bandpass filter (hmrBandpassFilt; high-pass = 0.01, low-pass = 0.50), was used to remove noise associated with physiological artefacts. Next, the OD data was converted to concentration changes using the modified Beer-Lambert Law (hmrOD2Conc; ppf = 6.0, 6.0) (*Delpy et al., 1988*). Lastly, the signal for each channel during the task condition was normalised to the preceding 5 s of the baseline (fixation) condition (hmrBlockAvg; trange = −5.0, 30.0). Subsequently, the normalised mean $[HbO_2]$ value between 5–30 s quantified $\Delta[HbO_2]$, and was used for analysis. The mean $[HbO_2]$ value was chosen as it has shown to be less impacted by artifacts (*Herold et al., 2018*) and the removal of the first and last 5 s after trial onset minimises confounding signals associated with physiological change, and anticipation of task cessation (*Jahani et al., 2018*). The mean $\Delta[HbO_2]$ of each channel was combined to form bilateral, left, and right DLPFC regions of interest (ROIs).

## Statistical analysis

Violations to assumptions of normality and homoscedasticity were evaluated using Q-Q plots prior to statistical analyses. Multiple univariate linear mixed models (LMMs) were used to quantify both behavioural changes (RT, accuracy) and neural activation. S-blocks and PA levels using METs (min/week) were set as fixed effects, with subjects as the random effect. All models were fitted using restricted maximum likelihood (REML) criteria. Satterthwaite's method to estimate degrees of freedom was applied. Goodness-of-fit tests were not conducted to avoid specification error, as all the fixed effect factors were needed to investigate the hypotheses.

For behavioural analysis, (1) mean RT of accurate trials only; and (2) accuracy proportion per S-block were used as the dependent variables (DVs). For changes in neural activation, $\Delta[HbO_2]$ bilateral, left, and right DLPFC hemispheres during S-blocks were used as the DVs. Alpha-value for significance was set at $p \leq 0.05$, and all statistical analyses were performed using R statistical software version 4.1.2 (R Project for Statistical Computing).

# RESULTS

## Participant characteristics

Participants had an average MET expenditure of 1,717.39 min/week. An average total time of 268.08-min was spent participating in PA of moderate intensity in a given week, with the total time spent on PA averaging to 314.04-min in a given week (Table 1).

## Behavioural results

As shown in Fig. 2A, results indicated a significant decrease in RT during S-blocks as METs increased ($\beta = -0.01$, $SE = 0.00$, $t = -2.26$, $p = 0.03$). There was also a significant decrease in RT from the first to last S-block ($\beta = -1.99$, $SE = 0.53$, $t = -3.74$, $p < 0.001$; Fig. 2B). For accuracy, 6 responses were deemed to be significant outliers, and removed from subsequent analysis. There were no significant effects of METs ($\beta = -0.00$, $SE = 0.00$, $t = -1.20$, $p = 0.24$) or block ($\beta = -0.00$, $SE = 0.00$, $t = -1.11$, $p = 0.27$) on accuracy across S-blocks.

## fNIRS

There was no significant effect of METs ($\beta = 0.00$, $SE = 0.00$, $t = 0.35$, $p = 0.72$) or block ($\beta = 0.02$, $SE = 0.12$, $t = 0.15$, $p = 0.88$) on bilateral $\Delta[HbO_2]$ across S-blocks (Figs. 3A, 3B). Likewise for analysis of left $\Delta[HbO_2]$ (Figs. 3C, 3D): METs ($\beta = -0.00$, $SE = 0.00$, $t = -0.18$, $p = 0.86$); block ($\beta = 0.02$, $SE = 0.12$, $t = 0.16$, $p = 0.88$); and right $\Delta[HbO_2]$ (Figs. 3E, 3F): METs ($\beta = 0.00$, $SE = 0.00$, $t = 0.64$, $p = 0.52$); block ($\beta = 0.02$, $SE = 0.18$, $t = 0.10$, $p = 0.92$).

# DISCUSSION

In this study, fNIRS was applied to examine whether PA levels, as defined by MET levels, impacted procedural motor skill acquisition and underlying neural activity in healthy young adults. To our knowledge, this was the first study to examine whether habitual PA influenced brain activity in regions of cognitive processing during procedural motor skill

**Table 1 Descriptive statistics.**

|  | N | Mean ± SD | Min | Max |
|---|---|---|---|---|
| Age (years) | 26 | 22.54 ± 1.68 | 21 | 27 |
| Sex | 13 M, 13 F | – | – | – |
| BMI | 26 | 21.69 ± 2.24 | 18.26 | 26.42 |
| Education (highest level) | 23 High School, 3 Bachelors | – | – | – |
| Physical activity (IPAQ) |  |  |  |  |
| Leisure activity (min) | 26 | 268.08 ± 317.89 | 0.00 | 1,350.00 |
| Total time (min) | 26 | 314.04 ± 353.15 | 0.00 | 1,370.00 |
| Total MET (MET-min/week) | 26 | 1,717.39 ± 1,594.58 | 0.00 | 5,769.50 |
| Time spent sitting per weekday (hrs) | 26 | 9.92 ± 3.36 | 4.00 | 16.00 |
| Time spent sitting per weekend (hrs) | 26 | 9.50 ± 3.70 | 4.00 | 16.00 |

**Note:**
N, number of participants; SD, standard deviation; IPAQ, International Physical Activity Questionnaire, MET, metabolic equivalent.

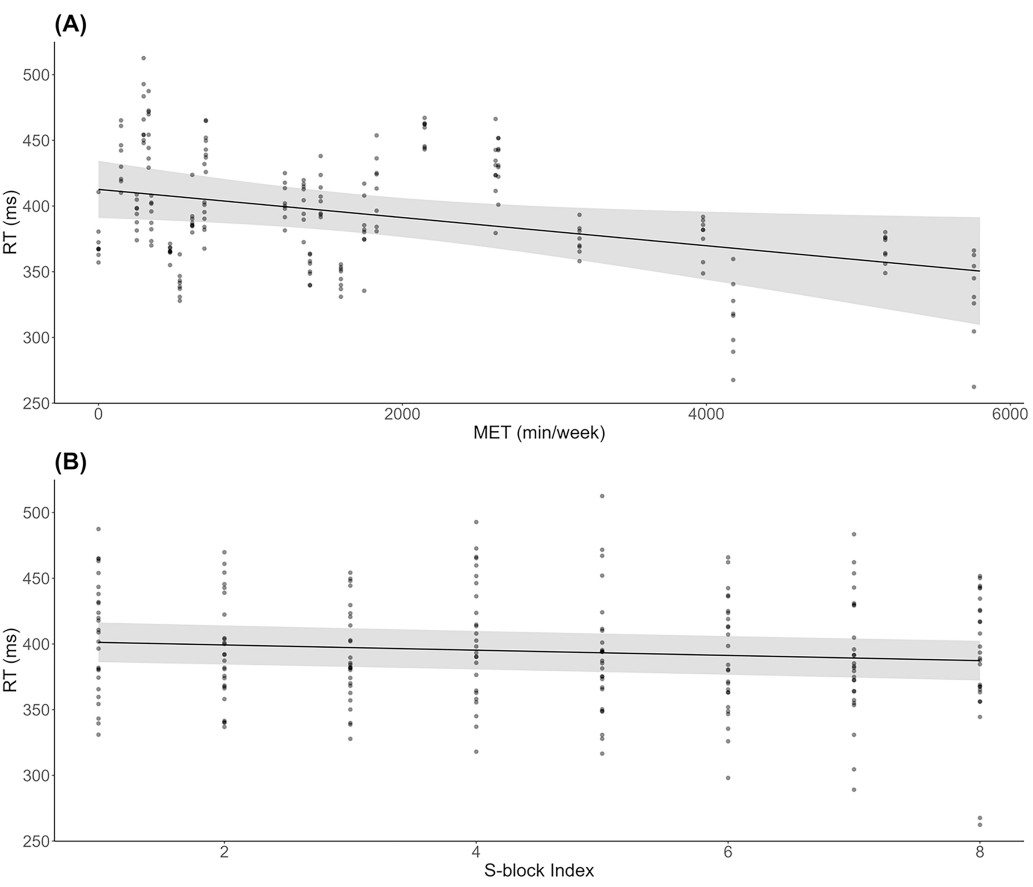

**Figure 2 Behavioural results across physical activity levels during sequence learning.** (A) Linear mixed model simple effect plot of reaction time (RT) across MET values; (B) linear mixed model simple effect plot of RT across S-blocks. Note: Each dot represents the RT value for each observation across MET values or blocks. The grey shading represents the 95% confidence interval for prediction from the linear model.

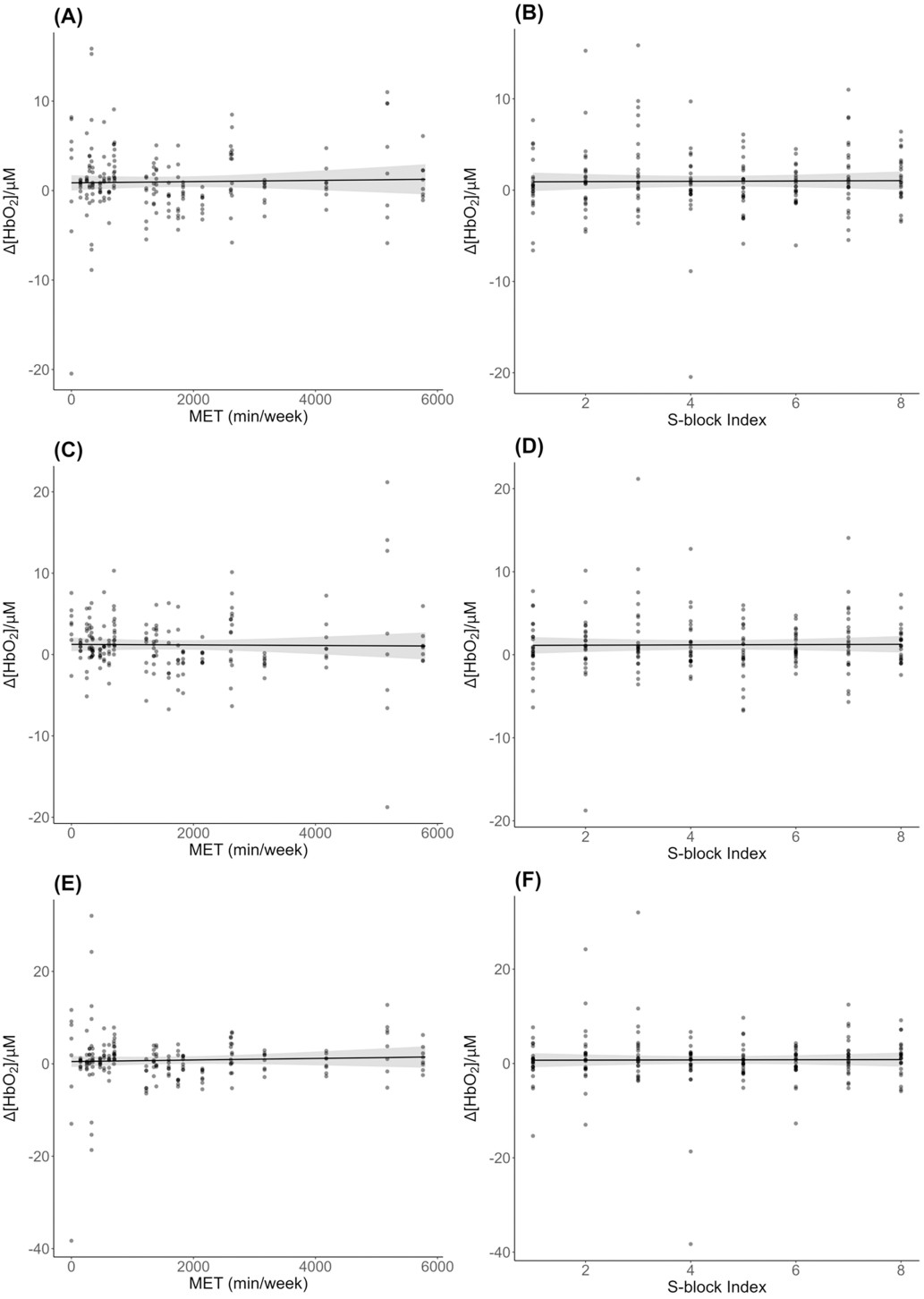

**Figure 3 DLPFC Δ[HbO₂] across physical activity levels during sequence learning.** (A) Simple effect plot of bilateral $\Delta[HbO_2]$ across MET values; (B) simple effect plot of bilateral $\Delta[HbO_2]$ across S-blocks; (C) simple effect plot of left hemisphere $\Delta[HbO_2]$ across MET values; (D) simple effect plot of left hemisphere $\Delta[HbO_2]$ across S-blocks; (E) simple effect plot of right hemisphere $\Delta[HbO_2]$ across MET values; (F) simple effect plot of right hemisphere $\Delta[HbO_2]$ across S-blocks. Note: Each dot represents the $\Delta[HbO_2]$ value for each observation across MET values or blocks. The grey shading represents the 95% confidence interval for prediction from the linear model.

acquisition. Our findings showed a significant decrease in RT as MET increased. However, there were no underlying changes in $\Delta[HbO_2]$ as MET increased. Overall, these findings have important implications for exercise prescription guidelines to optimise motor learning in younger, healthy populations. While the SRTT is an established paradigm to assess implicit motor learning (*Nissen & Bullemer, 1987*), our findings corroborated studies showing that increasing PA levels in young adults may be beneficial for cognitive processing and motor learning. Currently, our findings also underscore the need for sensitive task paradigms to mitigate ceiling effects in neural changes in otherwise healthy populations.

## Procedural motor skill acquisition

Previous research supports faster reaction times in physically active adults (*Spirduso, 1975*; *Lupinacci et al., 1993*; *Cirillo, Finch & Anson, 2017*); however, accounting for both reaction time and accuracy as measured in this study better reflects motor learning (*Nissen & Bullemer, 1987*). Results indicated no significant changes in accuracy as MET increased. One plausible explanation is that young, healthy adults experienced a ceiling effect earlier when acquiring a novel motor skill. On the other hand, observable behavioural differences in RT, as MET increased, could indicate that young adults with a more sedentary lifestyle may result in negative impacts on cognitive factors associated with procedural motor skill acquisition, such as working memory and processing speed (*Cirillo, Finch & Anson, 2017*). This finding contributes to the ongoing debate where studies have shown inconsistent results in establishing the relationship between PA and cognitive functioning in young adults (*Cirillo, Finch & Anson, 2017*). It is well established that observable age-related declines in both processing speed and working memory usually manifest around age 60 (*Dobbs & Rule, 1989*; *Salthouse, 2009*). This likely explains why PA levels can discriminate cognitive and/or motor performance among older compared with younger and middle-aged groups (*Spirduso, 1975*; *Cox et al., 2016*; *Zhu et al., 2023*), supporting recent findings suggesting both leisure-time PA and cardiovascular fitness predicted procedural motor skill learning in old, but not younger adults (*Zwingmann et al., 2021*). Therefore, our results aid in informing current literature that physical inactivity could have negative effects on motor skill acquisition in young, healthy individuals, which is consistent with findings within an older population.

## Dorsolateral prefrontal cortex (DLPFC) activation during procedural motor skill acquisition

Despite overall increases in DLPFC activation during procedural motor skill acquisition, no significant differences were observed as MET levels increased. The effect of PA and exercise on the function of the PFC in younger adults is inconclusive. In contrast to our results, previous findings from functional magnetic resonance imaging (fMRI) demonstrated increased DLPFC resting-state functional connectivity in overweight young adults, following a 6-month aerobic exercise intervention program (*Prehn et al., 2019*). This is in line with studies also citing enhanced DLPFC haemodynamic changes during

cognitive tasks in young adults after bouts of high-intensity exercise using fNIRS (*Kim et al., 2021*).

To date, research examining habitual PA levels and brain function has been largely limited to comparisons between older subjects who already may have underlying health conditions (*McGregor et al., 2013*; *Prehn et al., 2019*) and/or subjecting young, healthy participants to bouts of high-intensity exercise (*Kim et al., 2021*). In comparison, evidence from neural plasticity within the primary motor cortex (M1) has shown that continuous moderate-intensity exercise, which may be more reflective of leisure-like habitual PA may not be able to consistently enhance cognitive or motor performance and underlying brain plasticity compared to high-intensity exercise in healthy adults (*McGregor et al., 2013*; *Andrews et al., 2020*). Even in older adults, some findings have shown no differences in PFC structure or function between active and inactive groups (*Northey et al., 2020*; *Germain et al., 2023*). Moreover, age has always been seen as a mitigating factor in facilitating the level of cognitive arousal and brain function brought on by exercise (*Hayes, Forman & Verfaellie, 2016*). For example, during a motor inhibition task, PFC activity and reaction time were associated with PA level only in middle-aged, and older adults, but not in younger adults (*Berchicci, Lucci & Di Russo, 2013*). Despite the non-significant haemodynamic change between young adults of different PA levels in our study, behavioural results may be indicative of the negative effects of physical inactivity on cognitive functioning and motor skill acquisition. One potential explanation is that age may mediate the effect of habitual PA on brain function, where neurological differences may only be more apparent later in life (*Hötting & Röder, 2013*; *Cox et al., 2016*). Hence, it is imperative for future studies to delve into the longitudinal effects of PA on brain function and procedural motor skill acquisition across different life stages.

It could also be speculated that the improvements in reaction time during procedural motor skill acquisition may not solely driven by brain regions responsible for executive function, such as the DLPFC, and or any other region in isolation. Evidence have shown that various areas of cognition, such as visuospatial working and procedural memory, executive functioning, and processing speed, modulate the extent of procedural motor skill acquisition (*Daselaar et al., 2003*; *Aznárez-Sanado et al., 2022*). In this regard, it suggests that other brain areas may also be involved during the phase of motor sequence acquisition, where modulations in functional connectivity between and within various regions-of-interests (ROIs) during the procedural motor skill acquisition process may be more intrinsically linked to improvements in behavioural performance. In support of this, studies utilising fMRI have shown that various networks which included the parietal, ventrolateral, frontal, and medial premotor regions, were extensively recruited during procedural motor skill learning (*Daselaar et al., 2003*; *McGregor et al., 2009*; *Dayan & Cohen, 2011*; *Doyon et al., 2018*; *Aznárez-Sanado et al., 2022*). In addition, the sensorimotor lobule VI of the cerebellum have also demonstrated strong association with behavioural improvements, coupled with increased connectivity to various regions in the visual cortex during initial exposure to novel motor tasks (*Aznárez-Sanado et al., 2022*). Taken together, this may indicate that procedural motor skill acquisition as measured using SRTT might also involve various cortical and subcortical regions. This is evident

with similar research demonstrating no significant difference in activation in the left DLPFC during the follow-up session of a procedural finger tapping task when comparing participants who practiced the sequence *vs.* controls (*Polskaia et al., 2023*). The authors suggested that the DLPFC's function may be more catered to facilitating attentional direction during motor preparation, rather than movement execution when performing procedural motor skill acquisition tasks (*Jueptner et al., 1997*; *Polskaia et al., 2023*). This suggests that performance improvements during motor sequence acquisition may be facilitated by a dynamic and interconnected neural network, where the different brain substrates within the network contribute a unique functional aspect, rather than being solely modulated by a particular ROI (*Daselaar et al., 2003*). Hence, the recruitment and strengthening of connectivity within complex cognitive-motor neural networks involved in procedural motor skill acquisition may aid in the development of new visuospatial motor representations during learning, translating to better behavioural performance (*Aznárez-Sanado et al., 2022*). However, to gain a comprehensive understanding of the neural basis of procedural motor skill learning, future research should also consider the functional connectivity of various cognitive-motor networks in addition to investigating different brain regions in isolation.

## Limitations and future research

There are several limitations to this study. Firstly, despite power calculations, there was large variability in motor learning and associated neural responses. Future research should consider predictive factors such as age, sleep, and task complexity (*Luft & Buitrago, 2005*; *Taubert, Villringer & Lehmann, 2015*), which could help to explain individual variability in the association between PA, exercise, and motor learning. Moreover, it is possible that limitations using self-report PA may not accurately reflect habitual PA. Indeed, previous evidence has shown differences in objective and subjectively measured PA on cognitive/motor outcomes (*Cox et al., 2016*). Moreover, the benefits of PA on the brain and learning may evolve over longer periods of time, whereby PA levels are only measured as a snapshot of the preceding 1–2 weeks. Given the conflicting findings between the benefits of structured, laboratory-based exercise and habitual PA on brain activity and motor learning, the dose of exercise prescription, in particular the exercise intensity and type, may be important for optimising brain health and motor learning in otherwise younger, healthy populations. Whilst we aimed to capture moderate-to-vigorous type activities through the leisure section of the IPAQ, it is possible that a more targeted and personalised exercise prescription is needed to optimise brain health and learning in young, healthy populations (*Gourgouvelis et al., 2018*). Finally, while the SRTT is an established paradigm to assess implicit procedural motor learning in older adults (*Nissen & Bullemer, 1987*), future research should explore more complex and sensitive motor learning paradigms to capture the effect of PA on procedural motor skill acquisition in younger populations.

## CONCLUSION

The present study showed that there were no differences associated with brain activation in the DLPFC in young adults of different PA levels. Results indicated significant behavioural

differences, where greater habitual PA levels corresponded to faster RT. These findings corroborate current literature that PA levels may promote cognitive and motor performance to a different extent in young adults, with neurological differences only apparent later in life or only elicited *via* more demanding and complex motor learning tasks. Future research should explore the long-term effects of habitual PA on procedural motor skill acquisition and its related brain functions across the lifespan and under varying levels of task complexity in healthy populations.

## ACKNOWLEDGEMENTS

The authors wish to thank the study participants.

### Funding

This work was supported by the Nanyang Technological University Undergraduate Research Programme Project Consumables Fund (No. PESS21018). The funders had no role in study design, data collection and analysis, decision to publish, or preparation of the manuscript.

### Grant Disclosures

The following grant information was disclosed by the authors:
Nanyang Technological University Undergraduate Research Programme Project Consumables: PESS21018.

### Competing Interests

The authors declare that they have no competing interests.

### Author Contributions

- Fu-Miao Tan conceived and designed the experiments, performed the experiments, analyzed the data, prepared figures and/or tables, authored or reviewed drafts of the article, and approved the final draft.
- Wei-Peng Teo conceived and designed the experiments, authored or reviewed drafts of the article, supervision, and approved the final draft.
- Jessie Siew-Pin Leuk performed the experiments, authored or reviewed drafts of the article, and approved the final draft.
- Alicia M. Goodwill conceived and designed the experiments, analyzed the data, authored or reviewed drafts of the article, supervision, and approved the final draft.

### Human Ethics

The following information was supplied relating to ethical approvals (*i.e.*, approving body and any reference numbers):
Nanyang Technological University Institutional Review Board.

## Data Availability

The data is available at OSF: Tan, Fu Miao, Wei-Peng Teo, Jessie S P Leuk, and Alicia M Goodwill. 2024. "Effect of Habitual Physical Activity on Motor Performance and Prefrontal Cortex Activity during Implicit Motor Learning." OSF. April 16. doi: 10.17605/OSF.IO/TZPVK.

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
