# Peer review of "Effect of habitual physical activity on motor performance and prefrontal cortex activity during implicit motor learning"

_PeerJ, doi:10.7717/peerj.18217_

## Round 0.1 · original submission · Major Revisions

Heed the comments from both peer reviewers that require some major revisions

Reviewer 1 ·

Basic reporting

1. The description lacks clarity: In Figure 2, what do the dots represent?

2. Some statistical reporting is incomplete. For example, in Lines 232-233, the t-value, degrees of freedom, and p-value are missing. Additionally, it is confusing that the author states in Figure 2B that there were no significant effects of METs or block on accuracy across S-blocks, yet the figure actually shows RT vs. S-blocks.

3. It is unclear how the author derived delta[HbO2] from [HbO2].

Experimental design

1. Was delta[HbO2] averaged across both fixation and active task periods? To study the task-induced effect on neural activation, the power during the task period should be normalized by the power during the fixation period as a baseline.

Validity of the findings

1. The author claims an association between age and the benefit of PA on neural activation (Lines 38-40). However, this is not supported by any statistical evidence presented in the current study. This appears more speculative than a definitive result. I recommend moving this from the results/abstract section to the discussion.

·

Basic reporting

The article seems to fit well with the reporting requirements.
The authors clearly explain (1) the reason for the study and (2) relevant background information for the rationale for the task. This goes with inclusion of relevant references and context, the article being structured well (and, again, clearly). The hypothesis is clear and the results are discussed in the context of this.

The only minor comment in this regard is that a little more information about fNIRS and its utility for measuring neural activity would be nice (maybe just a sentence or so).

Experimental design

Like the basic reporting section, above, the article seems to fit well with the requirements in this area, describing the task and methods in terms of their use and previous relevance to the topic under consideration (notwithstanding the minor point where fNIRS is introduced).

Validity of the findings

The findings are generally clearly presented. The only comment relating to possible adjustments I would make is the clarity of Figures 2 and 3 - the text for the axes seems quite small for these and the meaning of the grey area seems to not have been included (although I may have missed it). Slight modifications would make these easier to examine by a reader.

The data is all available and the discussion is well expressed, both in terms of which it might mean, what the limitations are, and what would be beneficial to look at in future work.

Additional comments

There are a few (small) language errors, some of which are listed below (but a quick overall check would be good, too):

- L194: 'and Savitzky-Golay' -> 'and a Savitzky-Golay'
- L195: 'After which' is incorrect, 'After this' would be better.
- L198: 'using modified Beer-Lambert Law' -> 'using the modified Beer-Lambert Law'
- L223: 'Participants have' -> 'Participants had'
- L223: 'The average total' -> 'An average total'
Maybe the 'Behavioral' subtitle in the Results section would be better as 'Behavioural Results'
- L338: 'neural basis procedural' -> 'neural basis of procedural'
- L364-365: This sentence is either incomplete or doesn't need 'although' - 'The present study showed that although there were no differences associated with brain 365 activation in the DLPFC in young adults of different PA levels'

---

## Round 0.2 · accepted · Accept

Congratulations, your paper has been accepted.

Reviewer 1 ·

Basic reporting

no comment

Experimental design

no comment

Validity of the findings

no comment

·

Basic reporting

This all seems okay.

Experimental design

This all seems okay.

Validity of the findings

These all seem okay.

Additional comments

The authors have amended the manuscript to address the earlier review comments. I have no further comments to add and am happy to suggest acceptance of the paper.